# Polymorphic Characterization, Pharmacokinetics, and Anti-Inflammatory Activity of Ginsenoside Compound K Polymorphs

**DOI:** 10.3390/molecules26071983

**Published:** 2021-04-01

**Authors:** Yun-Yan Kuang, Xuan Gao, Yi-Jun Niu, Xun-Long Shi, Wei Zhou

**Affiliations:** 1Department of Chemistry, Fudan University, 2005 Songhu Road, Shanghai 200438, China; yykuang@fudan.edu.cn; 2Children’s Hospital, Fudan University, 399 Wanyuan Road, Shanghai 201102, China; naux1111@163.com; 3School of Pharmacy, Fudan University, 826 Zhangheng Road, Shanghai 201203, China; 20211030038@fudan.edu.cn (Y.-J.N.); xunlongshi@fudan.edu.cn (X.-L.S.)

**Keywords:** anti-inflammatory, ginsenoside compound K, pharmacokinetics, polymorph

## Abstract

Polymorphism exhibits different physicochemical properties, which can impact the bioavailability and bioactivity of solid drugs. This study focused on identifying the polymorphs of ginsenoside compound K (CK) and studying their different behaviors in pharmacokinetics (PK) and pharmacodynamics (PD). Four CK polymorphs (form I, II, III, and IV) from organic solvents were characterized by scanning electron microscope (SEM), differential scanning calorimetry (DSC), thermogravimetric analysis (TGA), Fourier transform infrared spectroscopy (FTIR), and powder X-ray diffraction (PXRD). A feasible LC-MS/MS method was exploited to determine the PK parameters. Form II displayed the most exposure, followed by form I, III, and IV. Notably, all forms showed sex dimorphism, and the bioavailability in the female group was about two-fold higher than in the male group. The PD properties were investigated in carrageenan-induced acute paw inflammation, and form II at 20 mg/kg showed significant inhibition of edema by 42.7%. This study clarified the polymorphic, PK, and PD characters of four crystal forms of CK, and the data suggested that form II had the best efficacy for drug development.

## 1. Introduction

Ginseng (*Panax ginseng*) is a well-known and prevalent traditional herb for its medicinal and healing properties in East Asian countries. Triterpene saponins, commonly known as ginsenosides, are the major pharmacologically active ginseng components [1,2].

20-*O*-β-D-glucopyranosyl-20(*S*)-protopanaxadiol, or ginsenoside compound K, is one of the major intestinal metabolites of ginsenoside Rb1, Rb2, and Rc after oral administration [3]. We have isolated a fungus of *Paecilomyces bainer* sp.229, which could effectively produce compound K (CK) by the biotransformation pathway of Rb1 → Rd → F2 → CK, and found a new approval of CK in treating rheumatoid arthritis (RA) [4,5,6]. It has been in phase I clinical trials in China as a novel oral candidate for RA therapy by Zhejiang Hisun Pharmaceutical Co., Ltd. (Zhejiang, China) [7].

In our previous studies, two single-crystal structures of CK solvates (dihydrate and methanol-water solvate) were reported [8,9]. Li et al. [10] further characterized another two solvates (nonstoichiometric hydrate and methanol solvate). Thus, the polymorphic phenomenon of CK is exhibited, but the influence of polymorphism on pharmacokinetics and pharmacodynamics is still unknown. Polymorphism is the ability of the same compound to be packaged in different ways in the solid state with different intermolecular interactions [11,12]. It exhibits different physicochemical properties [13,14] and can alter in vivo bioavailability, which finally modifies the drug efficacy [15,16]. Thus, appropriate analytical procedures to detect polymorphs and emphasize the importance of controlling the crystallization have great significance for the drugs administered in solid pharmaceutical dosages [17]. Therefore, it is crucial to pay attention to the solid-state forms at the beginning of CK development as an anti-RA candidate.

The objective of this study was to prepare and characterize different CK polymorphs by SEM, differential scanning calorimetry (DSC), TGA, FTIR, and powder X-ray diffraction (PXRD). Furthermore, an HPLC-MS/MS method with a negative electrospray ionization source and a rat paw swelling model induced by carrageenan was set up to evaluate the bioavailability and anti-inflammatory activity of these polymorphs.

## 2. Results

### 2.1. Scanning Electron Microscopy

Three techniques were used for the polymorphs preparation in the present study: (1) rapid solvent evaporation in a vacuum and high-temperature condition (form I); (2) gradual solvent evaporation in a small opening glass tube at room temperature (form II and IV), and (3) anti-solvent precipitation in water (form III) (see Section 4.3.1). During the solvent evaporation process, the solvents employed to affect the solute’s molecular aggregates presented in a supersaturated solution are a determinant of the final crystal form [18]. Here, we obtained four CK polymorphs from different polar solvents: form I from ethanol, form II from acetone, form III from methanol and anti-solvent precipitation of water, and form IV from methanol. These four crystal forms of CK exhibited different morphologies by scanning with a high-power electron microscope (Figure 1). Form I appeared amorphous solid, without a uniform and typical shape. Form II tended to clump together and presented an irregular granule structure. Form III tended to be scattered, and the crystal nucleus grew more evenly with thin flake form. In comparison, form IV showed a big block appearance and had a wavy surface under an optical microscope (Appendix A). These results indicated forms I, II, III, and IV displayed the apparent differences due to the different molecular arrangements in different solvents, respectively.

### 2.2. Thermal Analysis

The DSC thermograms and TGA traces of the four polymorphs are shown in Figure 2. For all the forms, TGA (Figure 2A inset) showed slight weight losses below 140 °C (less than 2.0%), indicating the presence of any solvated or hydrated forms, and these multi-step weight losses were attributed to the different interactions between different solvents and CK molecules. There was a pronounced weight loss of about 95% in TGA (Figure 2A) between 220 °C and 400 °C for every form, which corresponded to the CK molecule decomposition. However, there were noticeable shifts of the onsets in DSC curves (Figure 2B), 267.5 °C for form I, 263.1 °C for form II, 255.9 °C for form III, and 250.2 °C for form IV, which indicated the different thermodynamical stabilities of these forms.

Moreover, in DSC curves, form I exhibited only one endotherm peak of the decomposition. Form II had another two endothermal peaks at 115.4 °C and 186.6 °C, which were attributed to solvent desorption (acetone) and crystal melting, respectively. For form IV, the first solvent desorption (methanol) endothermal peak appeared at 120.1 °C, and the second melting endothermal peak followed at about 178.5 °C. As to form III, it had a broad peak from 50 to 100 °C, which was due to dehydration, and the weight loss was about 1% (Figure 2A). Meanwhile, it had no sharp endothermal melting peak as form II and form IV, so a phase transition might happen along with the solvate loss.

### 2.3. Fourier Transform Infrared Spectroscopy (FTIR)

The FTIR spectra (Figure 2C) were performed to confirm the results obtained from the thermal analysis, and the characteristic FTIR peaks were summarized in Table 1. The O-H stretching vibrations of the four forms had obvious shifts, at 3405 cm^−1^, 3362 cm^−1^, 3423 cm^−1^, and 3385 cm^−1^ for forms I, II, III, and IV, respectively (Figure 2C, dashed line). In form IV, there was another small peak at 3677 cm^−1^, which was the free O-H stretching vibration from the methanol molecules in this solvate, along with the bending vibration at 1308 cm^−1^. Nonpolar C-H stretching vibration and CH_3_/CH_2_ asymmetric deformation remained unchanged among the four forms. However, C=C stretching absorption peaks of form II and form III shifted to 1654 cm^−1^ and 1655 cm^−1^, respectively, a little higher than those of form I (1637 cm^−1^) and form IV (1639 cm^−1^). The weak bands at 1710 cm^−1^ and 1250 cm^−1^ were assigned to C=O stretching and bending vibration, which indicated the acetone molecules in form II. The fingerprint regions (600 to 1100 cm^−1^) of the four forms were very different in peak position and shape (Figure 2C, dashed line), which indicated the solvents interfered with the hydrogen bond formation between different glucosyl moieties of CK molecule in different ways.

### 2.4. Powder X-ray Diffraction (PXRD)

Figure 2D showed the PXRD data of the four forms, and the diffractograms indicated significant differences among them. There were only an extensive blunt band and no sharp peak on form I’s diffraction curve, which was considered an amorphous state. Form II and form IV showed the strongest peaks at 2θ of 14.60° and 14.89°, respectively, while the most substantial peak of form III appeared at a much lower 2θ angle of 6.84°. Furthermore, form II showed major peaks at 2θ values of 5.55°, 6.73°, 8.78°, 9.89°, 11.11°, 12.56°, 13.24°, 15.48°, 15.83°, 17.06°, 19.45°, 20.63°, 22.38°, 23.55°, 24.56°, 23.45°, 26.45°, and 30.99°. Form III exhibited major peaks at 2θ values of 9.47°, 10.54°, 12.48°, 13.69°, 15.32°, 17.17°, 21.62°, 23.12°, and 28.72°. Form IV had prominent peaks at 2θ values of 5.38°, 5.68°, 6.46°, 7.88°, 9.30°, 11.23°, 12.06°, 13.40°, 15.45°, 17.26°, 19.41°, 19.92°, 20.59°, 22.24°, 23.25°, 23.93°, and 29.85°. Thus, these different characteristic peaks and the results from DSC, TGA, and FTIR supported different crystal structures of the four forms.

### 2.5. In Vivo Studies: Pharmacokinetic and Pharmacodynamic Profiles

#### 2.5.1. Crystalline Polymorphic Impacts on Pharmacokinetic Properties

As shown in Table 2, the exposure to form IV was much lower than form I, and there was a 34% decrease in the bioavailability from form I to form IV. However, when the rats were treated with form II, a distinct increase in the area under the curve (AUC_(0–∞)_) was observed compared with the rats treated with form I. Thus, form II had the maximum bioavailability among the four forms, with 4.07% in the male group and 7.87% in the female group, respectively. The pharmacokinetics (PK) values of form III were just between form I and form IV. Taken together, although the PK parameters in rats treated with different forms did not show statistically significant differences (Appendix A), these results were of pronounced differences and indicated that the occurrence of CK polymorphs might alter its release step in the rat gastrointestinal fluids, reducing or increasing the extension of its absorption depending on the polymorphic form presented in the oral dosage.

#### 2.5.2. Sex-Related Impact on Pharmacokinetic Properties

It was found that females had higher exposure levels of CK than males in healthy Chinese subjects [19], so we also compared the PK properties of these CK polymorphs in sex. The results showed that no significant sex difference was observed in dose-normalized exposure parameters in this single-dose i.v. administration (Appendix A). While in p.o. administration, a very sex-related impact on the pharmacokinetic properties of all the four tested CK polymorphs was found (Table 2 and Figure 3). For every CK form, the average C_max_ and AUC in the female groups increased about two-fold compared to those in male groups. The AUC of form I and the C_max_ of form III had statistically significant sex differences according to one-way ANOVA analysis. Although without statistical significance, the average V_z/F_ and CL_z/F_ were also much more different between sex-groups in p.o. than those in i.v. However, T_max_ and t_1/2z_ showed no apparent sex dimorphism on any polymorph in both i.v. and p.o. administration.

#### 2.5.3. Crystalline Polymorphic Impacts on Anti-Inflammatory Properties

To further study the in vivo effects variation of different CK polymorphs, the pharmacodynamics was evaluated by measurements of time-dependent paw edema with carrageenan-induced acute inflammation (Figure 4). Compared with the normal group, edema reached the peak in the paw 3 h after carrageenan challenge with the maximum swell rate of 64% in the model group and then remitted gradually (Appendix A). The paw edema in the drug-treated groups showed the most significant inhibition at 5 h post-injection compared to the model group. Remarkably, form II at 20 mg/kg significantly reduced paw edema at 4, 5, and 6 h post-injection, with the inhibitions of 36.3%, 42.7%, and 40.0%, respectively.

In contrast, form IV displayed no statistical difference from the model group at any time point post carrageenan injection. Further, form II showed a statistical difference of paw edema from form IV at 4 h post-injection (39.7 ± 6.4% vs. 58.6 ± 3.6%, *p* = 0.034) at the dosage of 20 mg/kg. These results strongly suggested that the treatment with CK polymorphs in rats had the capacity to diminish the acute inflammation subsequently induced by carrageenan in vivo, and the anti-inflammatory effects were definitely related to the pharmacokinetic properties of the polymorphs after oral administration.

## 3. Discussion

Polymorphism is prevalent in the pharmaceutical field, which can impact the bioavailability of solid drugs [20,21]. CK was isolated and identified in 1972 [22], but data available on the polymorphism of CK were very limited [8,9,10]. Primarily, there is still no comparative study about the pharmacokinetic and pharmacodynamic properties of different CK polymorphs. In this study, four CK polymorphs were prepared and characterized, and significant differences were observed in crystal morphology. Form I was additionally proved as an amorphous state, with a halo pattern in PXRD and no melting peak in DSC [23]. The reason for the amorphous state of form I might be that it was produced from fast ethanol evaporation in diaphragm vacuum and high temperature (60 °C), thus the internal energy of the molecules was high, and the molecular movement was fast, which disturbed the formation the hydrogen-bonded aggregates of CK with the solvents for the growth of shaped crystals. Compared to form II and form III, PXRD peaks of form IV were more intense and sharp at high 2θ angles (20–30°), indicating that form IV had a better grain and a higher degree of crystallinity, which might occur due to the much slower crystallization rate. Form II peaks were less intense, indicating that form II probably had a more disordered structure and a discrete crystal size, as was observed in SEM (Figure 1). As to form III, there was a broad peak from approximately 50 to 100 °C, but no sharp melting peak at about 180 °C, which was different from form II and form IV. According to our previous single-crystal X-ray diffraction analysis [8], form III was originally presented to be a dihydrate. However, in this study, the weight loss of water in this TG analysis was only one-fifth of the stoichiometric value calculated for the dihydrate dehydration (Figure 2A). So, this form III was considered a novel anhydrous form of CK dihydrate by the water vaporization in the atmosphere during the storage or under dry nitrogen flow in DSC and TG analysis. Those having hydrogen bonding sites in the crystal can stoichiometrically incorporate water molecules into the crystal lattice of the drug and form a hydrate [24]. Conversely, the water molecules of a hydrate can be released from its crystal lattice under heating and drying conditions and transformed into anhydrous forms [25], even the amorphous form [26,27]. Thus, the detailed dehydration behaviors of form III deserved to be further investigated. Therefore, it was revealed that the four forms were different from each other and had different molecular arrangements in their crystal structures. By comprehensive comparison, form III was a novel anhydrous form of the dihydrate, while form I, form II, and form IV were almost but not the same as the reported forms [10,23,28].

The different crystal forms of an oral drug can affect its absorption and bioavailability and lead to differences in clinical efficacy [29,30]. However, to our knowledge, as an orally administrated drug, PK and pharmacodynamics (PD) studies of CK polymorphs have not been reported yet. Thus, in this manuscript, we conducted PK and PD properties of CK polymorphs for the first time. One of the important results, in oral administration, the sex dimorphism in the pharmacokinetics of these CK polymorphs was also observed in this study as in the previous report [7]. The maximum differences of the average AUC and C_max_ between male and female groups almost twice folded. In contrast, the differences in intravenous injection were tiny, indicating the process of absorption in the gastrointestinal tract influenced the sex dimorphism more than the metabolism in circulation. Over the past few years, researchers have gradually begun recognizing the importance of sex in drug development and clinical applications. However, the reasons behind the sex-related differences are yet not to be fully clarified [31,32]. Indeed, the sex-related expression of metabolism enzymes [33], transporters in target organs [34,35], and hormones secretion [36] were found to play a role in the differences between sexes. Nevertheless, additional factors, such as intestinal motility [37], food [19], and P-glycoprotein (P-gp) expression [38], could simultaneously be involved in altering the pharmacokinetics of CK.

Other than sex dimorphism, the variations of pharmacokinetics parameters among different CK polymorphs were also discovered as excepted. In our results, form II displayed the most potent bioavailable activity, followed by form I, III, and IV. The maximum differences of the average C_max_ and AUC_(0–∞)_ values compared to form II and IV occurred at 1.6-fold in male groups and 2.1-fold in female groups. Although there were remarkable differences, PK parameters did not present a statistically significant difference between genders [7,19], as well as among polymorphs. The reason was considered to the considerable individual variations of the CK pharmacokinetics; for example, C_max_ and AUC displayed significant variations resulting in standard deviation values, which were even close to half the value of the corresponding mean values. However, in the multiple-dose trial in humans [7], females had a significantly higher dose-normalized C_max_ and t_1/2_ than those in males after administration of CK on day 1 and day 15, which might be relevant to accumulation or hepatoenteral circulation in vivo after repeated administrations. More investigation about the absorption, distribution, metabolism, and excretion of CK needs in the future to explain this significant individual difference adequately.

Thirdly, in the classic carrageenan-induced rats’ paw edema, irrespective of sex, the paw edema levels decreased significantly compared to the model group, indicating the promising anti-inflammatory activity of CK reported previously [39,40]. The reduction of paw edema levels could be considered to be in good agreement with the pharmacokinetic profiles of the four CK polymorphs. Form II displayed the most potent anti-inflammatory activity, followed by form I, III, and IV. As a relative initial study, all of these results greatly enriched the PK and PD knowledge of CK polymorphs, which could provide beneficial information for this promising drug candidate’s crystal form selection. However, there were still some evident limitations: first of all, only four polymorphs were studied, and more crystal forms are likely to be discovered; secondly, more physicochemical properties, such as stability, dissolution rate, and solubility, will help explain the PK and PD differences; the thirdly, the novel form III presented the hydrate-anhydrous form transformation, and this transformation might occur under ambient conditions during drug processing, transportation, and storage. Evaluating the hydration and dehydration behaviors of form III is essential for developing a more stable formulation.

## 4. Materials and Methods

### 4.1. Materials

Raw CK was prepared in our lab [4] and higher than 98.0% in purity. Acetonitrile (HPLC grade) was purchased from Dikma Technologies Inc. (California, CA, USA). Ultrapure water used in the mobile phase was obtained by a Purist Pro water purification system (RephiLe Bioscience, Shanghai, China). All other reagents, solvents, chemicals, and solutions used were analytical reagent grades from Titan Scientific (Shanghai, China). Carrageenan (type IV) and indomethacin were purchased from Sigma-Aldrich (Missouri, MO, USA).

### 4.2. Animals

Sprague-Dawley rats (8 weeks old, 200 ± 30 g) were obtained from Shanghai SLAC Laboratory Animal Co., Ltd., (Shanghai, China). The animals were housed under a constant environment (temperature ~25 °C; humidity ~70%; and 12 h light-dark cycle) with free access to water and a rodent diet.

### 4.3. Methods

#### 4.3.1. Preparation of CK Polymorphs

Form I: Raw CK was dissolved in ethanol (10%, *w*/*v*) in a pear-shaped flask and vaporized to dryness on the rotavapor with a diaphragm vacuum at 60 °C in a water bath to obtain form I (amorphous solid).

Form II: Form I was dissolved in acetone (5%, *w*/*v*) in a glass tube (15 × 150 mm) at 60 °C in a water bath, and the solution was kept at room temperature overnight to obtain form II (granule-shaped). The tube was open for slow evaporation of acetone.

Form III: Form I was dissolved in methanol (10%, *w*/*v*) in a glass tube (15 × 150 mm) at 60 °C in a water bath, and distilled water was added to dilute methanol to 65% (*v*/*v*). The solution was kept at room temperature overnight to obtain form III (flake-shaped), with the tube was closed [8].

Form IV: Form I was dissolved in methanol (20%, *w*/*v*) in a glass tube (15 × 150 mm) at 60 °C in a water bath, and the solution was kept at room temperature for 4 days to obtain form IV (block-shaped), with the tube open for slow evaporation of methanol [9].

The four CK forms obtained from the experiment were air-dried at room temperature and then evaluated by various crystal characterization methods.

#### 4.3.2. Polymorphic Characterization

Scanning Electron Microscopy

The photomicrographs of the crystallized samples were examined morphologically with a SU8010 scanning electron microscope operating at 2.0 kV (Hitachi, Tokyo, Japan).

Thermal Analysis

DSC measurements were performed on a DSC204 Phoenix calorimeter (NETZSCH, Selb, Germany) with a heating rate of 20 K/min from 20 to 400 °C. The dry nitrogen flow was about 50 mL/min. Data were analyzed using NETZSCH analysis software.

TGA was obtained by a TGA8000 thermal analysis instrument (PerkinElmer, Waltham, MA, USA) at the same operating conditions as DSC, and data were analyzed using Universal Analysis software (TA Instruments, New Castle, DE, USA).

Powder X-ray Diffraction

PXRD data were collected at room temperature (25 °C) in the reflectance mode using a D2 PHASER diffractometer (Bruker AXS Inc., Madison, WI, USA) with Cu/K-α1 radiation (λ = 1.54056 Å) at 40 kV, 40 mA. Diffraction patterns (2θ) were collected from 3° to 40° at a step scan of 0.01° with a scanning speed of 0.6 sec. Results were analyzed using MDI/JADE 6.0 software (Materials Data, Livermore, CA, USA).

Fourier Transform Infrared Spectroscopy

The FTIR spectra of all samples were measured with a Nicolet iS10 spectrophotometer (ThermoFisher, Waltham, WI, USA). The samples were ground and mixed with potassium bromide (KBr) at 1% dilution, and the pellets were prepared by compressing the powder. The spectra were recorded in the range 400–4000 cm^−1^ and analyzed by Origin 7.5 software (OriginLab, Northampton, MA, USA).

### 4.4. Pharmacokinetic Studies

#### 4.4.1. Analytical Methods

All samples were analyzed with a triple quadrupole LC-MS/MS (API 4000 LC-MS/MS, AB Sciex, Framingham, MA, USA) coupled with an Agilent 1200 HPLC system (Agilent Technologies, Palo Alto, CA, USA). The MS was equipped with an electrospray system (ESI) [41]. The N_2_ gas temperature and ion-spray voltage were set to 550 °C and 4.5 kV, respectively. CK was measured in multiple reaction monitoring (MRM) mode using a negative electrospray ionization source at m/z 667.4 → 621.2 for CK (CID 25.6 v) and m/z 271.2 → 228.8 for internal standard (IS, (E)-5-(4-fluorostyryl)-2-isopropylbenzene-1,3-diol, Welichem Biotech Inc., Wayburne Drive Burnaby, BC, Canada) (CID 28.3 v) with a dwell time of 0.1 s.

Separation of CK was achieved with a Waters Atlantis C_18_ column (2.1 × 150 mm, 5 μm), and the elution gradient was defined as follows: A: actonitrile, B: 5 mM aqueous ammonium formate, 0~3 min: 40%A → 90%A, 4.19~4.2 min: 90%A → 40%A. The flow rate of the mobile phase was 0.35 mL/min, the injection volume was 10 μL, and the total run time was 6.4 min.

#### 4.4.2. Animals and Dosing

Rats were divided randomly into 5 groups of six animals each (sex in half) and given 7 days to acclimatize to the facility before the experiments began. The four forms of CK were suspended in 0.5% sodium carboxymethyl cellulose (CMC-Na) solution and orally administered to rats at a dose of 20 mg/kg. A CK solution in a mixed solvent of dimethyl sulfoxide (DMSO)/castor oil/physiologic saline (3:5:92, *v*/*v*) was intravenously administered to the remaining group via the tail vein at a dose of 10 mg/kg. Blood samples were collected by retro-orbital puncture at the following times: 0.167, 0.333, 0.667, 1, 2, 3, 4, 6, 9, 12, and 24 h for the oral administration, and at 0.0083, 0.05, 0.133, 0.25, 0.5, 1, 2, 4, 6, and 9 h for the intravenous administration. Plasma samples were obtained by centrifuging a heparinized blood tube at 4000 rpm for 5 min and then stored at −70 °C until analysis.

#### 4.4.3. Sample Preparation and Analysis

The procedure for drug extraction from plasma samples was as follows: 0.2 mL acetonitrile, 0.2 mL plasma sample, and 10 μL IS (1 µg/mL) were added into a 1.5 mL Eppendorf-type tube. This mixture was vortexed for 1 min and then treated with 0.8 mL of methyl tert-butyl ether (MTBE) to precipitate the plasma proteins. After a mechanical vortex for 1 min and subsequent centrifugation (4000 rpm, 5 min), 0.8 mL of the supernatant was transferred and blow-dried with N_2_ at 40 °C in a water bath. Finally, the residue was dissolved in 120 μL of mobile phase, and then 10 μL was subjected to HPLC for analysis. The assay was shown to be linear in the range 1.00–900 ng/mL with a linear regression equation of Y = 7.757 × 10^−3^ X + 1.885 × 10^−3^ (*r*^2^ = 0.999). Intra- and inter-day precision and accuracy never exceeded 15% by analyzing six replicate samples.

#### 4.4.4. Statistical Analysis

The CK pharmacokinetic parameters in rats, including the area under the curve (AUC_(0–t)_ and AUC_(0–∞)_), mean residence time (MRT_(0–t)_ and MRT_(0–∞)_), maximum concentration (C_max_), time to maximum concentration (T_max_), half-life (t_1/2_), apparent distribution volume (V), and total plasma clearance (CL), were derived from plasma concentration-time curves by using non-compartmental analysis (NCA) with DAS 2.0 software (Drug And Statistics, Mathematical Pharmacology Professional Committee of China, Shanghai, China). The absolute bioavailability (Fabs) was calculated from the dose-adjusted ratio of AUC_p.o._ to AUC_i.v._. One-way analysis of variance (ANOVA) by SPSS 11.5 (SPSS Inc., Chicago, IL, USA) was conducted to compare the primary PK parameters.

### 4.5. Anti-Inflammatory Tests

Rats were divided into 11 groups of six animals each (sex in half): the normal group, the model group, indomethacin group (1 mg/kg), and form I, II, III, IV groups (each form set 10 mg/kg and 20 mg/kg groups, respectively). Indomethacin and CK were ground and suspended in 0.5% CMC-Na solution. The rats were administered the drug orally for 3 days (qd), and those in the normal and model groups had the same volume of CMC-Na solution. One hour after the last administration, except for the normal group, 0.1 mL 0.1% carrageenan in sterile saline (0.9% NaCl) was injected into the plantar surface (i.pl.) of the right hind paw in order to induce inflammation. The same volume of saline was injected into the normal group. The volumes of the right hind paw of the rats were measured using a KW-7C plethysmometer (Nanjing KEW Bio-Technology, Nanjing, China) at different time intervals after injection (1, 2, 3, 4, 5, and 6 h). Paw edema and inhibitory rate were calculated as the following formula, and the differences among groups were compared with the *t*-test.

Paw edema (%) = (V_n_ − V_0_)/V_0_ × 100%.

V_n_: paw volume at different time intervals after carrageenan injection.

V_0_: paw volume before carrageenan injection.

Inhibitory rate (%) = (E_m_ − E_e_)/E_m_ × 100%.

E_m_: paw edema of the model group.

E_e_: paw edema of the experimental group.

## 5. Conclusions

Here, we prepared four different polymorphs of CK and fully demonstrated the polymorphism characteristics. Remarkable differences in PK and PD parameters of the polymorphs were observed after orally administered to rats. Form II showed a maximum bioavailability and therapeutic efficacy and could be preliminarily inferred as an advantageous polymorph. The present study can serve as a basic step to further polymorphic studies of ginsenoside compound K.

## Figures and Tables

**Figure 1 molecules-26-01983-f001:**
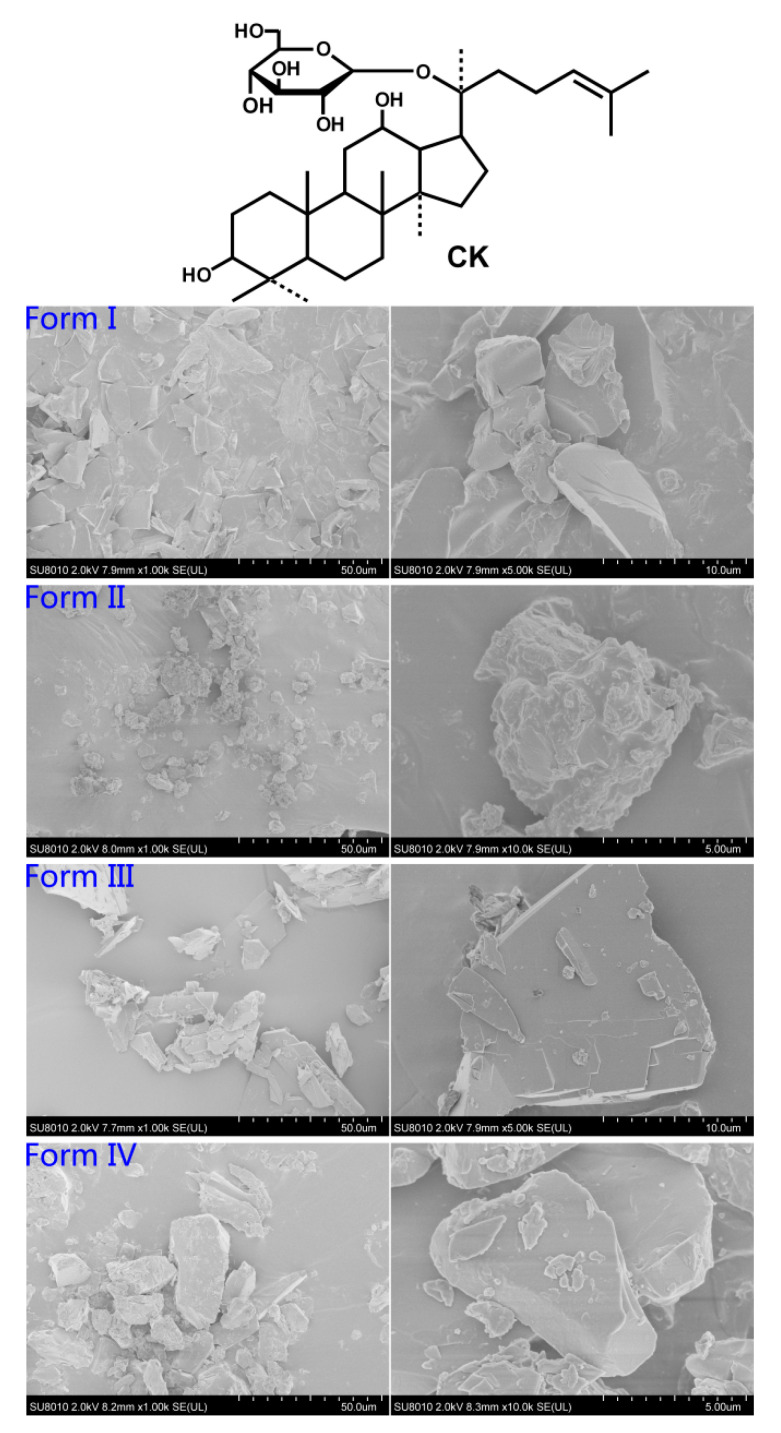
Chemical structure of compound K (CK) and scanning electron microphotographs of the four forms at 2.0 kV. Form I (magnification from left to right: 1000×, 5000×); form II (magnification from left to right: 1000×, 10,000×); form III (magnification from left to right: 1000×, 5000×); and form IV (magnification from left to right: 1000×, 10,000×).

**Figure 2 molecules-26-01983-f002:**
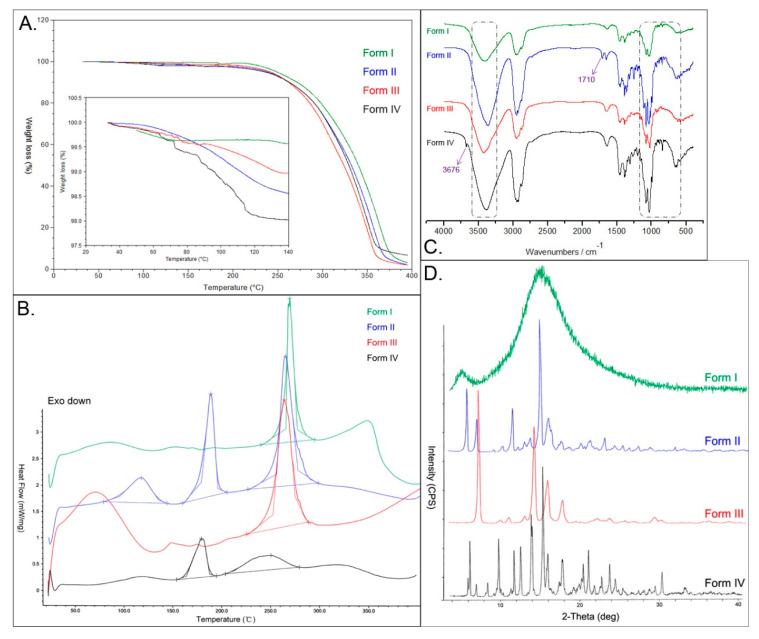
TGA (**A**) and differential scanning calorimetry (DSC) (**B**) curves of the four forms of CK were measured at 20 K/min from 20 to 400 °C under 50 mL/min dry nitrogen flow. DSC showed endothermic peaks of solvent desorption (below 140 °C), melting point (150–190 °C), and CK decomposition (250–270 °C). TGA showed a pronounced weight loss of about 95% between 220 and 400 °C for CK decomposition, and the inset showed a magnified view highlighting the weight loss blow 140 °C attributed to the solvent desorption. FTIR spectra (**C**) of the four forms of CK were from 400–4000 cm^−1,^ and the major changes observed in the spectra were marked with dashed lines and arrows. Powder X-ray diffraction (PXRD) patterns (**D**) of the four forms of CK were measured with Cu/K-α1 radiation (λ = 1.54056 Å) at 40 kV, 40 mA from 3° to 40° (2θ). Form I was in an amorphous state, and forms II, III, and IV were in crystalline structures.

**Figure 3 molecules-26-01983-f003:**
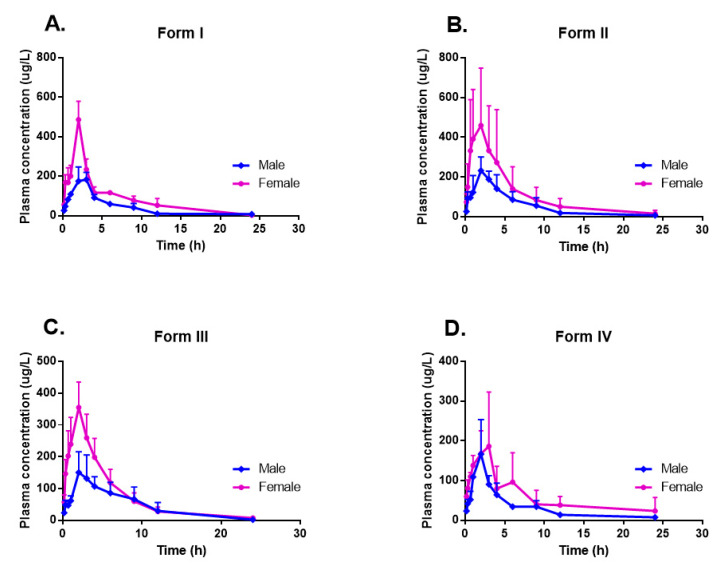
Plasma concentration-time profiles in sex groups after oral administration of the four forms of CK at a dose of 20 mg/kg in rats: (**A**) form I; (**B**) form II; (**C**) form III; and (**D**) form IV. Each point represents mean ± S.D. (*n* = 3).

**Figure 4 molecules-26-01983-f004:**
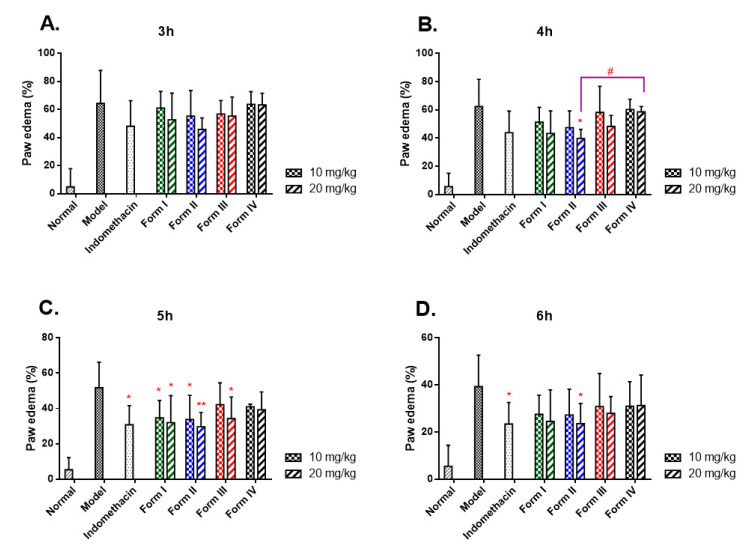
Results of paw swelling test induced by carrageenan of the four forms of CK at doses of 10 and 20 mg/kg in rats at different time intervals: (**A**) 3 h; (**B**) 4 h; (**C**) 5 h; and (**D**) 6 h. Values are expressed as mean ± SD of 6 rats in a group. *: *p*-value < 0.05, **: *p*-value < 0.01, versus the model group; #: *p*-value < 0.05, form II versus form IV (4 h, 20 mg/kg).

**Table 1 molecules-26-01983-t001:** FTIR spectra of specific bands of the four forms of CK.

	Forms	Form I	Form II	Form III	Form IV
Bands		Wavenumber (cm^−1^)			
Free O-H (methanol)					3677(*ν*), 1308(*δ*)
O-H stretching vibration		3405	3362	3423	3385
C-H stretching vibration		2876, 2944	2874, 2949	2876, 2948	2875, 2946
Free C=O (acetone)			1710 (ν), 1250 (δ)		
C=C stretching vibration		1637	1654	1655	1639
CH_3_, CH_2_ asymmetric deformation		1388, 1454	1390, 1456	1388, 1464	1388, 1454
C-O stretching vibration		990, 1037, 1076, 1118	991, 1013, 1035, 1076, 1111	989, 1025, 1077, 1095	990, 1031, 1078, 1111
C-C deformation		645	596	586	841, 632

Note: ν, stretching vibration; δ, bending vibration.

**Table 2 molecules-26-01983-t002:** Summary of pharmacokinetic parameters after oral administration of CK polymorphs in rats (20 mg/kg) (non-compartment model).

	Forms	Form I		Form II		Form III		Form IV	
Parameters		♂	♀	♂	♀	♂	♀	♂	♀
AUC_(0–t)_ (ug/L·h)		1149.8 ± 430.17	2202.7 ± 237.26 *	1191.1 ± 471.90	2447.2 ± 1828.5	1171.8 ± 462.65	1869.6 ± 618.65	758.62 ± 245.16	1312.0 ± 471.95
AUC_(0–∞)_ (ug/L·h)		1188.0 ± 480.37	2216.0 ± 225.74 *	1235.2 ± 474.45	2551.2 ± 1968.0	1183.5 ± 466.08	1896.2 ± 604.12	784.11 ± 219.63	1320.2 ± 471.22 #
t_1/2z_ (h)		3.0 ± 0.96	3.0 ± 0.84	2.9 ± 1.1	3.2 ± 0.75	3.3 ± 0.58	2.8 ± 0.25	3.5 ± 0.47	3.0 ± 0.60
T_max_ (h)		2.0 ± 0.0	2.0 ± 0.0	2.0 ± 0.0	2.0 ± 0.0	2.0 ± 0.0	2.0 ± 0.0	2.0 ± 0.0	2.3 ± 0.58
C_max_ (ug/L)		259.4 ± 144.2	453.3 ± 134.8	232.3 ± 68.97	459.6 ± 289.6	151.2 ± 64.45	355.1 ± 79.98 *	166.4 ± 87.20	218.8 ± 111.7
V_z/F_ (L/kg)		77.1 ± 28.3	40.5 ± 14.2	74.1 ± 31.6	45.3 ± 20.0	93.6 ± 50.1	44.8 ± 9.37	136 ± 43.2	73.6 ± 37.7
CL_z/F_ (L/h/kg)		18.7 ± 7.27	9.09 ± 0.880	18.0 ± 7.11	10.8 ± 5.79	19.4 ± 9.46	11.2 ± 3.02	26.8 ± 6.91	16.5 ± 5.67
Fabs (%)		3.92%	6.84%	4.07%	7.87%	3.90%	5.85%	2.59%	4.07%

Note: AUC_(0–t)_: area under the plasma concentration-time curve from zero to the time of the last quantifiable concentration; AUC_(0–∞)_: area under plasma concentration-time curve from zero to infinity; t_1/2z_: half-life; T_max_: time to maximum plasma concentration; C_max_: maximum plasma concentration; V_z/F_: apparent volume of distribution after administration; CL_z/F_: the total plasma clearance of the drug after administration. All values are presented as mean ± SD (*n* = 3). The significant effect of sex was investigated by one-way ANOVA for the pharmacokinetics (PK) parameters. ♂: male rats; ♀: female rats. *****: *p*-value < 0.05 versus the male group. **#**: *p*-value < 0.05 versus form I.

## Data Availability

Data are contained within the article or Appendix A.

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
