# Peer review of "Polymorphic Characterization, Pharmacokinetics, and Anti-Inflammatory Activity of Ginsenoside Compound K Polymorphs"

_molecules, 2021, doi:10.3390/molecules26071983_

Round 1

Reviewer 1 Report

The submitted manuscript presents the results of the studies on the four polymorphic forms of CK. This compound has been proven to be an important drug candidate. I think that this study is worth publishing after major revision. My detailed comments can be found below. Besides, the level of English should be greatly improved. There are many language mistakes that really diminish such an interesting study.

The Abstract is too long, too detailed and there are many not needed phrases such as “for the first time” or “These findings held great significance in the research of CK polymorphs.”

Line 14, CK should be defined.

Line 17, PK should be defined.

Line 18, PD should be defined.

Line 20, please delete “Four different forms(Form I, II, III,and IV)of CK polymorphs were identified”, it was already mentioned earlier in the Abstract

Line 32, not “healthy” but “healing”

The Authors should create of Figure with a structural formula of CK.

[17] is very old, try to cite a newer publication that focuses on the comparison of the analytical methods in the analysis of the solid drug forms, there are plenty of examples.

Line 58, why not SCXRD?

Lines 82 and 105 are the same.

Line 97, it shouldn’t be “dehydration of water”; term “dehydration” means the loss of water, so there can’t be any other type of dehydration

Line 120, 121-do you suggest that form IV might be a hydrate? Or non-stoichiometric hydrate? It should be calculated, based on the molecular weights of CK and single water molecule, either to confirm or deny this hypothesis.

Line 133, not “spectrum” but “spectra”.

Table 2, the data are not rounded correctly. For the uncertainties the maximum number of significant figures should be two.

FT-IR, more information are needed. Howe were the samples prepared? Using ATR? Or pellets with KBr?

PXRD results should be used to determine at least the unit cell dimensions and crystal space groups, i.e. using Pawley refinement. It can be easily and quickly done using various friendly software.

Line 393, it should be “statistical analysis” not “statistics analysis”

Author Response

Dear referee,

Thank you very much for your peer-review. We appreciate these useful comments and suggestions, and have revised the manuscript accordingly. Attached please find the revised manuscript for your review. The revised parts are highlighted with yellow background.

The detailed corrections are listed below point by point:

1, “Besides, the level of English should be greatly improved. There are many language mistakes that really diminish such an interesting study.”

Ö Thanks for the suggestion. We’ve read the manuscript carefully and tried our best to correct those language mistakes. All the corrections are highlighted with yellow background.

2, The Abstract is too long, too detailed and there are many not needed phrases such as “for the first time” or “These findings held great significance in the research of CK polymorphs.”

Ö Thanks. As suggested, the Abstract was simplified and rephrased. The phrases of “for the first time” and “These findings held great significance in the research of CK polymorphs.” were deleted.

3, Line 14, CK should be defined.

Ö Thanks for the suggestion. The abbreviation was defined.

4, Line 17, PK should be defined.

Ö Thanks for the suggestion. The abbreviation was defined.

5, Line 18, PD should be defined.

Ö Thanks for the suggestion. The abbreviation was defined.

6, Line 20, please delete “Four different forms(Form I, II, III,and IV)of CK polymorphs were identified”, it was already mentioned earlier in the Abstract.

Ö Thanks for the suggestion. In the revised Abstract, this duplication was deleted.

7, Line 32, not “healthy” but “healing”

Ö Thanks for the suggestion. We’ve corrected it.

8, The Authors should create of Figure with a structural formula of CK.

Ö Thanks for the suggestion. The structural formula of CK was added in Figure 1 of the revised manuscript.

9, [17] is very old, try to cite a newer publication that focuses on the comparison of the analytical methods in the analysis of the solid drug forms, there are plenty of examples.

Ö Thanks for the suggestion. A newer publication was cited. “Novakovic, D.; Isomaki, A.; Pleunis, B.; Fraser-Miller, S.J.;  Peltonen, L.; Laaksonen, T.; Strachan, C.J. Understanding dissolution and crystallization with imaging: A surface point of view. Mol. Pharmaceutics 2018, 15, 5361-5373.”

10, Line 58, why not SCXRD?

Ö Thanks. Form I was amorphous solid and Form II was opaque, so they were not suitable for SCXRD analysis. Form III and Form IV were transparent in crystallization solutions and the SCXRDs were reported in our previous studies [see Ref 8 & 9]. But they became un-transparent after drying in air or long-time storage and then were not suitable for SCXRD analysis any more. In this study, all the forms were air-dried at room temperature, and then were used for characterization, PK, and PD studies. So SCXRD analysis could not be conducted.

11, Lines 82 and 105 are the same.

Ö Thanks for the suggestion. The duplication was deleted.

12, Line 97, it shouldn’t be “dehydration of water”; term “dehydration” means the loss of water, so there can’t be any other type of dehydration

Ö Thanks for the suggestion. It was corrected.

13, Line 120, 121-do you suggest that form IV might be a hydrate? Or non-stoichiometric hydrate? It should be calculated, based on the molecular weights of CK and single water molecule, either to confirm or deny this hypothesis.

Ö Thanks. Do you mean Form III? According to our previous SCXRD analysis [see Ref 8], Form III presented to be a dihydrate. In this study, the weight loss of water in TG analysis was only one-fifth of the stoichiometric value calculated for the dihydrate. So we thought it became a non-stoichiometric hydrate (an anhydrous form) and this content was discussed in the section of Discussion (Line 208-224).

14, Line 133, not “spectrum” but “spectra”.

Ö Thanks for the suggestion. It was corrected.

15, Table 2, the data are not rounded correctly. For the uncertainties the maximum number of significant figures should be two.

Ö Thanks. But we couldn’t catch the meaning of this suggestion very well. According to our limited understanding, we rounded the data again and the significant figures’ number of mean value and SD value were kept the same.

16, FT-IR, more information are needed. Howe were the samples prepared? Using ATR? Or pellets with KBr?

Ö Thanks for the suggestion. The samples were prepared with KBr, and the general of experimental section was added in the revised manuscript.

17, PXRD results should be used to determine at least the unit cell dimensions and crystal space groups, i.e. using Pawley refinement. It can be easily and quickly done using various friendly software.

Thanks. To be honest, we are not good at the skill of PXRD analysis and unable to determine the unit cell dimensions or crystal space groups by using a refinement method. But we think, in our further studies of investigating the stability, dissolution rate, solubility, dehydration, and crystallization behaviors of these polymorphs, we will seek the technical help from other experts to give more information about the unit cell dimensions or crystal space groups.

18, Line 393, it should be “statistical analysis” not “statistics analysis”

Ö Thanks for the suggestion. It was corrected.

Thanks for your consideration, and looking forward to your kind response.

Best wishes,

Sincerely

Wei Zhou

Reviewer 2 Report

  1. The abstract is written very technical. However, please rephrase it as it must state the aim, brief methodology and approach, results and in the last sentence the conclusion.
  2. In the abstract: please give full definition to any name before making abbreviation,
  3. Paragraph 55-62: please rephrase the paragraph and star with a rationale and objective of the study.
  4. the conclusion paragraph is too descriptive in detail and please move most of them to discussion paragraphs and summarize the conclusion into one short paragraph as readers get the conclusion far quickly.
  5. In line 166-176, please start the paragraph with the specific aim and approach sought in measuring plasma concentration.

Author Response

Dear referee,

Thank you very much for your peer-review. We appreciate these useful comments and suggestions, and have revised the manuscript accordingly. Attached please find the revised manuscript for your review. The revised parts are highlighted with yellow background.

The detailed corrections are listed below point by point:

1, The abstract is written very technical. However, please rephrase it as it must state the aim, brief methodology and approach, results and in the last sentence the conclusion.

Ö Thanks. As suggested, the Abstract was rephrased, and the aim, brief methodology, results and the conclusion were all stated.

2, In the abstract: please give full definition to any name before making abbreviation

Ö Thanks for the suggestion. All the abbreviations in the Abstract were defined.

3, Paragraph 55-62: please rephrase the paragraph and star with a rationale and objective of the study.

Ö Thanks. As suggested, the paragraph was rephrased and the objective of this study was presented at the beginning.

4, the conclusion paragraph is too descriptive in detail and please move most of them to discussion paragraphs and summarize the conclusion into one short paragraph as readers get the conclusion far quickly.

Ö Thanks for the suggestion. The conclusion paragraph was rephrased and simplified. Only the brief conclusion was presented.

5, In line 166-176, please start the paragraph with the specific aim and approach sought in measuring plasma concentration.

Ö Thanks for the suggestion. We exchanged the sequence of 2.5.1 and 2.5.2 sections for a better narrative logic and started the paragraph with the specific aim.

Thanks for your consideration, and looking forward to your kind response.

Best wishes,

Sincerely

Wei Zhou

Round 2

Reviewer 1 Report

The Authors have honestly answered to my questions which I highly appreciate. Though the manuscript is not perfect, it is suitable for publication.